# Comprehensive Research on Druggable Proteins: From PSSM to Pre-Trained Language Models

**DOI:** 10.3390/ijms25084507

**Published:** 2024-04-19

**Authors:** Hongkang Chu, Taigang Liu

**Affiliations:** College of Information Technology, Shanghai Ocean University, Shanghai 201306, China; 2152416@st.shou.edu.cn

**Keywords:** ESM-2, PSSM, druggable protein, deep learning, machine learning

## Abstract

Identification of druggable proteins can greatly reduce the cost of discovering new potential drugs. Traditional experimental approaches to exploring these proteins are often costly, slow, and labor-intensive, making them impractical for large-scale research. In response, recent decades have seen a rise in computational methods. These alternatives support drug discovery by creating advanced predictive models. In this study, we proposed a fast and precise classifier for the identification of druggable proteins using a protein language model (PLM) with fine-tuned evolutionary scale modeling 2 (ESM-2) embeddings, achieving 95.11% accuracy on the benchmark dataset. Furthermore, we made a careful comparison to examine the predictive abilities of ESM-2 embeddings and position-specific scoring matrix (PSSM) features by using the same classifiers. The results suggest that ESM-2 embeddings outperformed PSSM features in terms of accuracy and efficiency. Recognizing the potential of language models, we also developed an end-to-end model based on the generative pre-trained transformers 2 (GPT-2) with modifications. To our knowledge, this is the first time a large language model (LLM) GPT-2 has been deployed for the recognition of druggable proteins. Additionally, a more up-to-date dataset, known as Pharos, was adopted to further validate the performance of the proposed model.

## 1. Introduction

Druggability refers to the potential of a target to be effectively influenced by pharmaceutical interventions. Meanwhile, the notion of the “druggable genome” [1] was initially introduced in 2002, characterized by a protein’s potential to interact with a modulator and yield a desired therapeutic outcome. They joined forces to coin the term “druggable protein”, which can be used in a narrow sense to describe a protein’s capability to interact with small molecule ligands that modify its function or in a broader sense to signify a protein’s ability as a viable therapeutic target in the treatment of human diseases. Druggable proteins play a crucial role in drug development and discovery. Yet, assessing the druggability of a potential target protein is a complex and challenging task with no universally accepted solution [2].

Traditionally, the primary methods for identifying novel druggable proteins have centered on biochemistry techniques, encompassing approaches like pocket estimation methods [3], the detection of protein-protein interaction sites [4,5], and ligand-specific methods [6]. However, these technical approaches face obstacles such as low efficiency, high costs, and a substantial dependence on advanced equipment and skilled personnel. Additionally, a protein target found to be undruggable late in the drug discovery process represents a significant expenditure of time and resources within the pharmaceutical industry [7]. Time-saving and good efficiency are crucial in the prescreening process. To overcome these limitations, computational strategies have been applied with the help of large amounts of data [8]. 

Machine learning methods have been used and continue to yield better results with the surging protein sequence databases in distinguishing druggable proteins from undruggable ones. In 2012, support vector machine (SVM) and random forest (RF) algorithms were employed by Yu et al. [5], utilizing data sourced from the DrugBank database [9]. In 2016, Jamali et al. pioneered the use of a neural network classifier for predicting druggable proteins; physicochemical properties, amino acids, and dipeptides were chosen as features [10]. This approach attained a cross-validation accuracy of 92.10%. They developed a balanced dataset that has been extensively utilized in subsequent studies. Lin et al., in 2018, extracted protein features using dipeptide composition, pseudo-amino acid composition, and reduced sequence, followed by a genetic algorithm for feature selection [11]. They improved the SVM classifier with bagging ensemble learning for prediction, achieving an accuracy of 93.78%. In 2022, Yu et al. implemented a hybrid deep learning model with dictionary encoding, dipeptide composition, tripeptide composition, and composition-transition-distribution, achieving an accuracy of 92.40% and a recall of 94.50% (higher than Lin et al.) [12]. 

Many researchers have employed machine learning and deep learning techniques, including neural networks [10,12], SVM [13], RF [5], etc. [14]. However, within the realm of bioinformatics, natural language processing (NLP) models are increasingly gaining researchers’ interest [15]. Word embeddings are widely utilized for NLP tasks to capture semantic characteristics and linguistic relationships within text data, converting raw text into numerical vectors or metrics that facilitate the development of machine learning (ML) models [16]. Notably, protein sequences bear a resemblance to natural languages in human beings, as they are constructed from different amino acids, effectively forming a ‘language of life’ [17]. Recently, a new generation of deep-learning-based language models has emerged, which is designed to generate embeddings for protein sequences. These models, such as ESM-2 [18], UniRep [19], and ProtTrans [17], are trained on extensive protein sequence datasets and excel at creating informative protein representations solely from sequence information. 

In this paper, we employed evolutionary scale modeling 2 (ESM-2) for the analysis of druggable proteins across several well-established datasets, including DrugBank [9], Pharos [20], and those used in other papers. The data we used is the pure protein sequence. When applying traditional machine learning algorithms, we used ESM-2 to generate 320-dimensional vectors and then input them into classifiers like SVM, naive Bayes (NB) [21], and extreme gradient boosting (XGB) [22] using the Python scikit-learn library [23]. We also incorporated deep learning techniques such as capsule networks (CapsNets) and bidirectional long short-term memory (BiLSTM) [24] networks, conducting extensive training over many epochs to optimize parameters. For instance, in the case of CapsNets, we explored various kernel sizes to determine the optimal settings. To maximize ESM-2’s capabilities, we fine-tuned it by training on Jamali’s dataset [10]. This fine-tuning process sought to produce better 320-dimensional embeddings, which are aimed at enhancing performance metrics for distinguishing druggable proteins within the same predictive model framework. To broaden the scope of our research, we also investigated features through non-machine-learning techniques, including a position-specific scoring matrix (PSSM) using the Blast software (v2.13.0) [25]. Our analysis revealed that pre-trained models marginally surpass PSSM-based features in terms of both embedding efficiency and overall accuracy. Moreover, we presented a novel methodology by leveraging a language model originally intended for purposes other than protein-related tasks to evaluate its efficacy in druggable protein classification. This marked the first instance of directly employing a modified generative pre-trained transformer 2 (GPT-2) [26] language model for classification problems related to druggable proteins. Figure 1 outlines the main framework of our research, tracing the pathway from protein sequence data through a series of predictive models, culminating in the identification of druggable proteins, as described in the preceding text.

## 2. Results and Discussion

### 2.1. Machine Learning Classifiers Performance

We utilized 5-fold cross-validation (CV) to guarantee the selection of the most effective model and to prevent overfitting. Detailed information regarding the division of our dataset into training, validation, and test sets is comprehensively outlined in Section 3.1. Following this, Table 1 assessed the performance of traditional classifiers on Jamali’s dataset, while Table 2 addressed the Pharos dataset. The terms accuracy, precision, sensitivity, specificity, F1 Score, and Matthews Correlation Coefficient are abbreviated as ACC, P, SN, SP, F1, and MCC, respectively. The corresponding receiver operating characteristic (ROC) curves are shown in Figure 2. The Shapley Additive Explanations (SHAP) [27] analysis of XGB, SVM, and RF for Jamali’s dataset is shown in Figure 3. We calculate the average magnitude of SHAP values for each feature across all samples by first taking the absolute values and then averaging these. Specifically, we sum the absolute SHAP values for the S-FPSSM, DPC-PSSM, and KSB-PSSM feature groups (each containing 400 features) and then compute the average importance of these groups. This approach quantifies the impact of each feature group on the models’ predictions. Through further calculation, it is found that S-FPSSM exhibits the highest mean absolute SHAP value, making it the most important PSSM feature, followed by DPC-PSSM and KSB-PSSM. For S-PSSM, higher values strongly correlate with a protein being druggable, as seen from the placement and color of the dots. Conversely, lower values decrease the likelihood. The magnitude of a SHAP value indicates the strength of a feature’s influence on the model’s decision.

Comparatively, the ESM-2 encoding method generally resulted in better classifier performance across both datasets, with SVM and XGB classifiers frequently achieving the highest scores in various metrics. The best accuracy on Jamali’s dataset was achieved with ESM-2 encoding and an SVM classifier, reaching 0.9326. For the Pharos dataset, the highest accuracy was obtained using ESM-2 embedding with an XGB classifier, achieving an accuracy of 0.8739. In the SHAP value analysis, S-FPSSM was the top feature influencing the model’s predictions, ranking above DPC-PSSM and KSB-PSSM in terms of combined feature importance.

### 2.2. Deep Learning Classifiers Performance

In this study, we extended our research to deep learning models on 5-fold CV, diverging from previous works that did not employ neural network-based classifiers [11]. The training was conducted on Jamali’s dataset and Pharos dataset, employing DNNs, CapsNets, and BiLSTM networks. Table 3 and Table 4 show the results of the two datasets. The ESM-2 encoding consistently outperformed PSSM across both Jamali and Pharos datasets, leading to higher performance with various classifiers. Specifically, CaspNet with ESM-2 encoding achieved the highest accuracy on Jamali’s dataset (0.9340), while DNN with ESM-2 encoding showed superior performance on the Pharos dataset (0.9037). This indicates the effectiveness and generalizability of ESM-2 encoding in enhancing classifier outcomes despite its lower dimensionality compared to PSSM.

For the CapsNets, we trained them with different kernel sizes since they achieved the best results on Jamali’s dataset. The evaluation of the ACC and MCC on the independent test set is shown in Figure 4. This plot facilitated an understanding of the impact that different kernel sizes and epochs have on the model’s performance, revealing that a kernel size of three was the most effective for this dataset.

In Figure 5, we employed the Uniform Manifold Approximation and Projection (UMAP) [28] to visualize the abstract features learned by different neural network architectures from high-dimensional data. This data was extracted from the second-to-last layer of each model, which typically contains the most meaningful representations for classification tasks, just prior to the final decision-making layer. In the plot below, 1 represented druggable, and 0 represented undruggable. Two datasets (Jamali’s dataset and Pharos), two different features (1200-dimensional PSSM features and 320-dimensional esm2_t6_8M_UR50D embeddings), and three kinds of neural networks (BiLSTM, CapsNet, DNN) were used, resulting in 12 plots. The purpose of these visualizations is to compare how different architectures organize and differentiate the data based on the learned features. Observations from these plots reveal distinct patterns of data clustering, which are significantly influenced by the choice of embedding technique and the nature of the dataset used. These patterns provide insights into the effectiveness of each architecture in distinguishing between druggable and undruggable proteins, thereby guiding the selection of the most suitable model and features for the identification.

### 2.3. Performance Variance Analysis

In the case of the Pharos dataset, which was used in its pure form without being integrated with other datasets, there were no existing benchmark comparisons. Thus, we opted to compare it with Jamali’s dataset. When comparing the ACC of the Pharos dataset to Jamali’s dataset, using identical types of features and classifiers, it became apparent that Jamali’s dataset yielded slightly better results. Two reasons accounted for the differences in our findings. Firstly, the reduced size of the Pharos dataset, intended to create a balanced dataset, resulted in the elimination of many protein sequences. Despite containing over 20,000 protein sequences, the Pharos dataset suffered from a lack of balance, leaving only 704 sequences that could be regarded as reliably druggable. This was significantly less than Jamali’s dataset, which had over 1200 druggable proteins. The diminished size of Pharos’s training set could adversely affect the model’s learning capabilities. Another key difference lay in the labeling variance between the datasets. For example, 43 proteins were classified differently in terms of druggability in the Pharos dataset and Jamali’s dataset, with 37 marked as druggable in Pharos but not in Jamali’s, and vice versa for 6 (Figure 6). This discrepancy could be attributed to the advancements in the pharmaceutical industry, given that the Pharos dataset was relatively more recent. Furthermore, the labeling attributes in both datasets differed. Jamali’s dataset contained labels marked as undruggable that were classified as Tchem in Pharos, whereas Tchem labels indicate the ability to bind small molecules with high potency [29]. It’s noteworthy that numerous Tchem entities, which could potentially be classified as Tclin (druggable), were currently categorized as undruggable. This was a common issue in the classification of druggable proteins. As the field progresses, many proteins initially deemed negative in our training sets could turn out to be positive. These three factors likely contributed to the variance observed between the two datasets.

### 2.4. Comparison with State-of-the-Art Methods

In our study, we evaluated our models against earlier classifiers using Jamali’s dataset. Table 5 displays the comparison results, with all computational findings sourced from the papers that introduced these models. Our most effective model emerged as the fine-tuned ESM-2 using Jamali’s dataset with CapsNet. We also drew the protein contact maps of our fine-tuned model (Figure 7). The visualizations are matrix (generated by ESM-2) plots where each element of the matrix corresponds to a pair of amino acids in the protein sequence. The color intensity or shading in each cell of the matrix indicates the likelihood or strength of interaction between these amino acid pairs. It can be observed that the contact map generated by our fine-tuned model, shown in the second row, focuses on specific areas. This suggests that the model is well-trained. Notably, the model excelled in speed, requiring only a few minutes to predict a sequence-based protein dataset when running on platforms like Google Colab or Kaggle. This efficiency is a significant improvement over traditional methods that rely on handcrafted features, making our approach more effective in feature generation and prediction. The modified GPT-2 model does not achieve very good results; however, as an end-to-end model, it does not require any feature generation tasks.

### 2.5. PSSM Versus PLM

PSSM excels in pinpointing protein domains and functional sites by using evolutionary data. In contrast, the protein language model (PLM) broadens this scope by using large datasets to predict various biological properties, including structure, function, and interactions. This approach complements the detailed insights gained from PSSM with extensive predictive capabilities. In the comparative analysis of classification tasks in this study, it has been observed that fine-tuning embeddings from the ESM-2 model demonstrate marginally superior performance over PSSM features. However, this outcome could be significantly influenced by the specific datasets employed. The selection of varied PSSM features and the choice of different pre-trained models are factors that might also affect these results. Notwithstanding, when considering practicality, pre-trained models hold an advantage due to their time efficiency and effective utilization of computational resources, particularly GPU power. This contrasts with the PSSM approach, which can be notably time-intensive as it iterates through the entire dataset using CPU.

### 2.6. Web Server

Previous research has significantly advanced our understanding of druggable proteins. To make the latest findings accessible, some researchers have developed web servers. For example, Jamali et al. established a web server (https://www.drugminer.org/ (accessed on 19 March 2024)) that enables users to explore druggable proteins identified in their research, along with the comprehensive dataset utilized. Similarly, Arwa Raies et al. introduced a web application (http://drugnomeai.public.cgr.astrazeneca.com (accessed on 19 March 2024)) that allows for the visualization of druggability predictions and essential features that determine gene druggability, categorized by disease type and modality [32]. Inspired by these state-of-the-art efforts, we have launched a web server available at https://www.druggableprotein.com (accessed on 19 March 2024). Current web servers related to druggable proteins are shown in Table 6. The web application interface is shown in Figure 8. Running on a computer with 1 core and 2 GB of RAM, this platform offers users the capability to upload FASTA files for prediction, as well as to generate fine-tuned ESM-2 embeddings and 1200D PSSM embeddings.

## 3. Materials and Methods

### 3.1. Datasets

In our research, we worked with two datasets for the analysis. Initially, we picked the DrugMiner dataset [10], a well-regarded benchmark created by Jamali in 2016. This is quite a balanced dataset. It includes 1224 druggable proteins and 1319 undruggable proteins. The positive samples, which are proteins approved as drug targets by the FDA and correspond to a variety of diseases like leukemia, thrombocytopenia, angina pectoris, and hypertension, were sourced from the DrugBank database [9]. On the other hand, the negative samples, proteins that cannot be considered drug targets, were collected from Swiss-Prot, employing the methods proposed by Li et al. [33] and Bakheet et al. [34]. Additionally, we used a highly imbalanced dataset sourced from Pharos [20]. This original dataset consists of four classes: Tbio (12,277) [35], Tchem (1915), Tdark (5516), and Tclin (704) [36], with 20,142 protein sequences in total. Among these classes, we only used two of them, Tclin and Tdark. Tclin is recognized as the druggable category. Tdark comprises unstudied proteins, which are mainly considered undruggable. However, it may still contain some proteins with potential druggability.

We kept Jamali’s dataset in its entirety, which was also adopted by many other research papers [11,12,14,37]. For the Pharos dataset, we encountered a distinction where “Tdark” represented the unstudied proteins, while “Tbio” and “Tchem” exhibited varying degrees of druggability. To address this ambiguity, we conducted two methods on this dataset.

Method 1: We began by selecting all the “Tclin” terms (704 in total) and subsequently employed random sampling from the “Tdark” dataset while maintaining result reproducibility by Python’s random seed (set to 42) to form the final dataset (704 out of 5516). This approach was adopted to ensure a balanced dataset, resulting in 704 positive samples (druggable proteins) and 704 negative samples (undruggable proteins).

Method 2: We directly designated “Tclin” terms as positive data, while the remaining terms were considered negative. To maintain dataset balance, we applied the SMOTE (Synthetic Minority Over-sampling Technique) method [38], resulting in 11,432 positive samples (druggable proteins) and 11,432 negative samples (undruggable proteins).

Although Method 2 achieved a high accuracy during the training process, it underperformed in the validation or test stages. Consequently, Method 1 was employed for the Pharos dataset in this study. 

We divided the Jamali’s dataset and Pharos dataset into training and testing sets with an 80:20 ratio. Within the training set, we further split the data into 80% for training and 20% for validation. Table 7 details the number of samples used in each process. Table 8 provides a simple description of the range of protein sequence lengths and other statistical information, including the mean and standard deviation rounded to integer values for clarity.

### 3.2. Feature Representation

#### 3.2.1. PLM Embeddings

A PLM can create an embedding of consistent size for proteins of any length. Among all sorts of language models, the ESM-2 model class was chosen for its speed and effectiveness [19]. The ESM-2 language models were trained using the masked language modeling objective, which involves predicting the identity of amino acids randomly selected within a protein sequence, relying on their contextual information within the remainder of the sequence [18]. The training approach enables the model to capture dependencies between amino acids. The model can generate a numerical vector of 1280 dimensions (esm2_t33_650M_UR50D) or 640 dimensions (esm2_t12_35M_UR50D) or 320 dimensions (esm2_t6_8M_UR50D) for each protein. For our research, we selected the esm2_t6_8M_UR50D model, which produces a full output as a dictionary that includes logits for language modeling predictions, attention weights, and contact predictions. In our specific downstream classification task, we extracted the sixth representation layer of this model, corresponding to the final layer, which yielded a 320-dimensional vector output. Additionally, the attention contacts produced by the esm2_t6_8M_UR50D model were utilized to draw protein contact maps [39]. These maps are crucial for identifying potential targets for drug binding. To elucidate, a protein contact map offers a simplified representation of a protein’s three-dimensional structure. It systematically records the proximity of amino acids within the protein, with each contact point in the map representing a pair of amino acids that are close together in three-dimensional space, typically within a predetermined threshold distance. This 2D matrix representation, unlike the full atomic coordinates, remains invariant regardless of the protein’s orientation and position. Such a characteristic makes contact maps particularly suitable for analysis via computational models, enabling simpler and more effective predictions by machine learning techniques due to their reduced complexity compared to complete 3D structures.

#### 3.2.2. PSSM Features

PSSM is often used to establish evolutionary patterns for extracting features. The PSSM generated by Blast [25] is represented as a matrix with dimensions L*20, where L denotes the protein sequence’s length. In this study, all the PSSM matrices were produced by employing PSI-BLAST against the SWISS-PROT database [40]. This searching process involves three iterations, using an E-value cutoff of 0.001 for multiple sequence alignment. We concatenated three PSSM-based features, DPC-PSSM (400D), KSB-PSSM (400D), and S-FPSSM (400D), to form a 1200-dimensional vector for each protein. The details about the three PSSM features are shown below.

DPC-PSSM: By extending the traditional dipeptide composition (DPC) from the primary sequence to incorporate the PSSM, a new method termed DPC-PSSM has been created [41]. This approach is designed to reflect the influence of local sequence order. It calculates the frequency of each possible dipeptide (a pair of adjacent amino acids) in a protein sequence. With 20 standard amino acids in existence, there are 400 (20 × 20) possible dipeptides. The 400-dimensional vector can be defined as:(1)$$Y={\left(y_{1,1},\ldots ,y_{1,20},\ldots y_{2,20},\ldots ,y_{20,1},\ldots y_{20,20}\right)}^{T},$$
(2)$$y_{i,j}=\frac{1}{L-1}\sum_{k=1}^{L-1}p_{k,i}\times p_{k+1,j}\left(1\leq i,j\leq 20\right).$$

KSB-PSSM: K-Separated-Bigrams-PSSM builds upon the concept of dipeptide composition, focusing on pairs of amino acids that are distanced by ‘k’ intervening amino acids [42]. By setting a key parameter k (we used k = 3), it calculates the transfer probability between amino acids separated by two others, resulting in a 400-dimensional feature. The calculation is shown below:(3)$$Y={\left(y_{1,1}\left(k\right),\ldots ,y_{1,20}\left(k\right),y_{2,1}\left(k\right),\ldots ,y_{2,20}\left(k\right),\ldots ,y_{20,1},\ldots ,y_{20,20}\left(k\right)\right)}^{T},$$
(4)$$y_{i,j}\left(k\right)=\sum_{t=1}^{L-k}p_{t,i}\times p_{t+k,j}\;(1\leq i,j\leq 20).$$

S-FPSSM: Derived from FPSSM by row transformation, S-FPSSM is a 400-dimensional feature [43]. FPSSM itself is a matrix that filters out negative values in the original PSSM. The calculation involves summing products of elements in the FPSSM and a delta function indicating the presence of a specific amino acid at a given position. The formula is calculated as:(5)$$Y_{j}^{\left(i\right)}=\sum_{k=1}^{L}fp_{k,j}\times \delta_{k,i},$$
(6)$$\left\{\begin{array}{c} \delta_{k,i}=1\;\mathrm{if}\;r_{k}=a_{i}\\ \delta_{k,i}=0\;\mathrm{otherwise} \end{array}(1\leq i,j\leq 20)\right..$$

In the given equation, $fp_{k,j}$ is the value in the FPSSM’s kth row and jth column, $r_{k}$ is the kth amino acid in the protein sequence, and $a_{i}$ is the ith ranked amino acid in the PSSM. These three PSSM-based features enhance the classifier’s performance by fully capturing the evolutionary information in protein sequences. We conducted a SHAP analysis on the importance of three PSSM-based features using XGB as the classifier, following the explanatory framework introduced by Lundberg et al. [27]. SHAP leverages game theory principles to quantitatively evaluate the influence of various input features on prediction outcomes, which assess the average marginal contribution of each feature across all possible combinations, thus ensuring a fair attribution of each feature’s impact on model predictions.

### 3.3. Model Architecture

#### 3.3.1. Machine Learning Methods

In this paper, we compared the classification performance of various traditional machine learning classifiers, such as SVM, RF, NB, and XGB, on selected features. We used the scikit-learn package (v1.3.2) [23] to implement all these models. We employed the grid search method for hyperparameters optimization, and Table 9 shows our final settings using the Python scikit-learn library.

#### 3.3.2. DNN

The deep neural network (DNN) architecture provides the foundational structure for deep learning. Rooted in the principles of artificial neural networks (ANNs), a traditional machine learning algorithm, DNNs have been instrumental in advancing various research and application fields. The key distinctions between DNNs and ANNs lie in the depth of hidden layers, the connections between layers, and the capability to learn features effectively across diverse datasets. Essentially, a DNN is an evolved version of a multilayer neural network featuring multiple hidden layers situated between the input and output layers, often referred to as a multilayer perceptron (MLP) [44]. We devised a DNN featuring three hidden layers with 180, 60, and 30 neurons, respectively, culminating in an output layer with a single neuron. Each layer employed the ReLU activation function to introduce non-linearity alongside a dropout strategy with a rate of 0.5 to prevent overfitting. The model uses binary cross entropy as the loss function, is optimized with the Adam optimizer at a learning rate of 0.001, and processes data in batches of 10. Figure 9 shows the DNN architecture.

#### 3.3.3. CapsNet

CapsNets [45], which are variants of convolutional neural networks (CNNs) [46], have emerged as a novel approach in the field of deep learning. Unlike CNNs, CapsNets are structured to recognize hierarchical relationships within data, which is highly beneficial in understanding complex structures like biological sequences. Like CNNs, CapsNets have convolution layers, padding, and strides but do not have pooling layers, which are used to reduce the spatial dimensions of the representation. Pooling, typically employed to reduce the dimensions of data representations in these networks, often results in the loss of essential spatial information. This loss is particularly problematic in biological sequences where the detailed spatial relationships between various high-level features are critical for accurate analysis and interpretation. 

In a CapsNet, the primary structural unit is the ‘capsule’, a group of neurons whose activity vector represents the instantiation parameters of a specific type of entity, such as an object or an object part. The length of this activity vector represents the probability that the entity exists, and its orientation encodes the instantiation parameters. This is a significant shift from the scalar-output feature detectors of CNNs, allowing for a more dynamic and interpretable approach. The model employs binary cross entropy as the loss function, uses the Adam optimizer with a learning rate of 0.001, and convolutional layers with a kernel size of 3 and a stride of 1. Figure 10 shows the CapsNet architecture.

#### 3.3.4. BiLSTM

Long Short-Term Memory (LSTM) networks [47] are specialized subsets of Recurrent Neural Networks (RNNs) designed to solve common issues like the vanishing gradient problem found in traditional RNNs. They utilize a system of input, forget, and output gates to control the flow of information, enabling them to selectively retain or discard information over extended sequences. Building on the gating mechanism, BiLSTM [24] networks further refine the LSTM networks by processing data in both forward and reverse directions through two parallel LSTM layers. 

Our BiLSTM model configuration included an input dimension of 320, a hidden dimension of 64, and an output dimension of 2. This model processes data in batches of 16 and utilizes the Adam optimizer with a learning rate of 0.001 to optimize the model parameters, incorporating binary cross entropy as the loss function. The architecture took advantage of a bidirectional strategy to analyze sequences through an LSTM layer that processes information in both forward and backward directions. This method allowed for a comprehensive understanding of input features from both perspectives, significantly enhancing prediction accuracy. Figure 11 shows the BiLSTM architecture. 

### 3.4. LLM Solution: Modified GPT-2

ESM-2 is a transformer-based model designed for PLMs, but many large language models (LLMs) already exist [45]. These LLMs should also be capable of doing downstream tasks, just like the ESM-2 model. Recognizing the potential of such models in tasks beyond NLP, we adapted the GPT-2 [26] architecture to classify proteins as druggable or undruggable. 

The original GPT-2 vocab size was 50,257; we changed it to 33, which matches the alphabet size of ESM-2 models. We reduced the block size, number of heads, and layers to accelerate prediction speed. The input to the model was the tokenized protein sequence, and the output consisted of classification probabilities rather than embeddings, rendering this task end-to-end. Essentially, we fed a pure protein sequence into the model, and it returned classification results, indicating whether the sequence was druggable or undruggable.

We modified the GPT-2 model, which is written in Pytorch framework (https://github.com/karpathy/nanoGPT (accessed on 19 March 2024)). The model started by encoding the protein sequence into a unified representation, where the final embedding was obtained by summing position embeddings and token embeddings (Figure 12a). These embeddings were then transformed through a series of self-attention and feed-forward neural network layers (transformer blocks), and finally, a softmax function was applied to generate probabilities for the two classes (Figure 12b). The modified architecture introduced a novel aspect by adapting GPT-2, initially designed for language understanding, to effectively capture intricate patterns in protein data.

### 3.5. The Fine-Tuning Process

In the fine-tuning process, we trained the esm2_t6_8M_UR50D model with some additional layers on Jamali’s dataset. This process was aimed at generating better 320-dimensional embeddings.

Starting from the encoding of the protein sequences, we employed an alphabet comprising 33 characters, which is the same as the original ESM-2 models. This alphabet included a unique ‘X’ symbol representing proteins absent in nature, alongside specialized tokens designated for specific functions: ‘cls’ for classification, ‘unk’ for unknown sequences, ‘eos’ for marking the end of sequences, and ‘pad’ for sequence padding. A protein of length L is encoded into a one-dimensional vector, also of length L, where each element ranges from integer values of 0 to 33. This is achieved through dictionary encoding using the specified alphabet. To address computational limitations, the contact prediction head in our model utilizes an input feature size of 24, substantially reduced from the 120 features used in the standard ESM-2 model. We ensured a balanced dataset by aligning and equalizing the number of positive and negative samples. Following the approach used in the ESM-2 embedding with DNN architecture, we incorporated a fully connected layer and a classification layer in our model. When using 320-dimensional embeddings for downstream tasks, we extract the results before the fully connected layer and the classification layer.

During the training process, proteins were initially sorted based on their sequence length. Subsequently, a batch size of two was employed to ensure uniform padding to a consistent length. Opting for a smaller batch size proves advantageous as it minimizes excessive padding, mitigating information loss. The chosen loss function was binary cross-entropy (BCE). Additionally, we analyzed the unsupervised self-attention map contact predictions [47] in the representation layer to understand the attention allocation between druggable and undruggable proteins. Attention contacts allow the model to focus on different parts of the input sequence when producing a particular part of the output sequence. This mechanism is analogous to how human attention works when we focus on certain aspects of a visual scene or a piece of text.

### 3.6. Performance Evaluation

Our assessment of the predictive models for distinguishing druggable proteins involved 5-fold CV and independent tests. By averaging the performance across different data subsets, we ensured the model’s robustness and minimized overfitting. This method allowed us to select the most consistent and reliable model across all folds. To evaluate the performance of the proposed model, six standard metrics were utilized: accuracy (ACC), precision (P), sensitivity (SN), specificity (SP), F1, and Matthews correlation coefficient (MCC). These metrics were derived from the counts of true negatives (TN), true positives (TP), false positives (FP), and false negatives (FN). Detailed calculations are shown below.
(7)$$ACC=\frac{TP+TN}{TP+FP+TN+FN}$$
(8)$$P=\frac{TP}{TP+FP}$$
(9)$$SN=\frac{TP}{TP+FN}$$
(10)$$SP=\frac{TN}{TN+FP}$$
(11)$$F1=2\times \frac{TP}{2TP+FP+FN}$$
(12)$$MCC=\frac{\left(TP\times TN\right)-\left(FP\times FN\right)}{\sqrt{\left(TP+FN\right)\times \left(TN+FP\right)\times \left(TP+FP\right)\times \left(TN+FN\right)}}$$

Furthermore, we employed ROC curves and the area under the curve (AUC) to illustrate the model’s capability. The AUC value, which ranges from 0 to 1, serves as an indicator of performance quality, with 1 denoting the best performance and 0 indicating the worst. Notably, an AUC of 0.5 signifies performance equivalent to random prediction.

## 4. Conclusions

In this study, we focused on identifying druggable proteins using two different datasets. We explored various methods to create input features, particularly using the PLMs and PSSMs, with a range of models, including machine learning classifiers (SVM, NB, RF, XGB) and deep learning classifiers (DNN, CapsNet, BiLSTM). The ESM-2 model performed better than PSSM features in our tests on two datasets. This highlights the strength of PLMs in predicting druggable proteins, suggesting that PLMs might offer an advantage over traditional search methods. However, it’s important to consider that these results might be influenced by the specific characteristics of the datasets used. The performance of PLMs could also vary depending on the dataset context.

Crucially, our work underscores the transformative potential of PLMs in the realm of protein classification through the deployment and fine-tuning of models like ESM-2. These models, with their ability to generate robust protein feature embeddings, represent a promising frontier for future research in drug discovery and protein science. Furthermore, although the modified GPT-2 model did not achieve the best performance in our current setup, its inclusion marks a pioneering step towards integrating LLMs in predicting druggable proteins. The flexibility and generalizability of PLMs and LLMs suggest vast potential for broader applications in the field, and they may shed light on other protein-related annotation tasks. The corresponding source code can be found at https://github.com/txz32102/DruggableProtein (accessed on 19 March 2024), and the web server is available at https://www.druggableprotein.com (accessed on 19 March 2024).

## Figures and Tables

**Figure 1 ijms-25-04507-f001:**
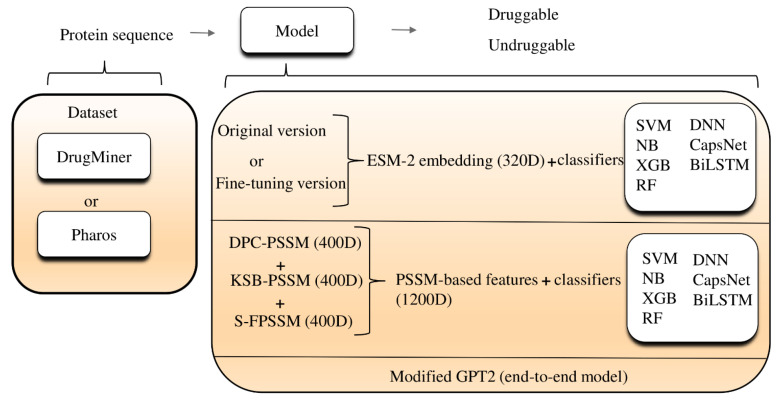
Flow chart of our study. ESM-2: evolutionary scale modeling 2; 320D: 320 dimensions; SVM: support vector machine; DNN: deep neural network; NB: naive bayes; XGB: extreme gradient boosting; CapsNet: capsule networks; BiLSTM: bidirectional long short-term memory; RF: random forest; DPC-PSSM: dipeptide composition position-specific scoring matrix; KSB-PSSM: K-Separated-Bigrams position-specific scoring matrix; GPT2: generative pre-trained transformer 2; 400D: 400 dimensions; 1200D: 1200 dimensions.

**Figure 2 ijms-25-04507-f002:**
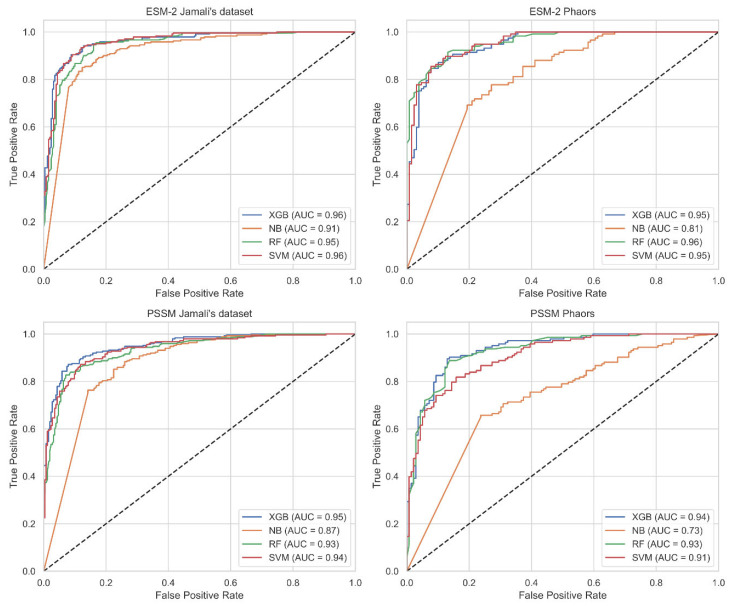
ROC curves for different datasets using various features with a 5-fold CV on the test set. Abbreviations: SVM: support vector machine; XGB: extreme gradient boosting; NB: naive bayes; RF: random forest; AUC: accuracy; ESM-2: evolutionary scale modeling 2; ROC: receiver operating characteristic; CV: cross-validation.

**Figure 3 ijms-25-04507-f003:**
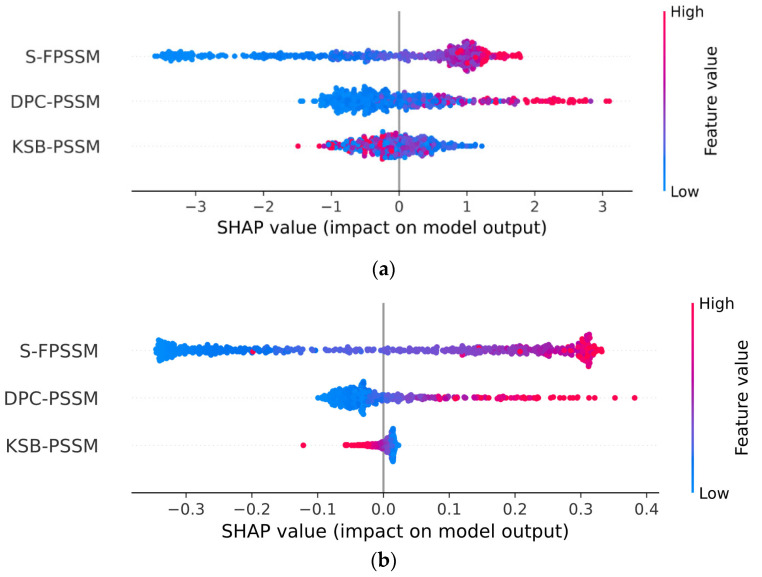
SHAP analysis of the importance of PSSM-based features on Jamali’s dataset. (**a**) XGB; (**b**) SVM; (**c**) RF. Abbreviations: SHAP: Shapley Additive Explanations; XGB: extreme gradient boosting; SVM: support vector machine; RF: random forest; DPC-PSSM: dipeptide composition position-specific scoring matrix; KSB-PSSM: K-Separated-Bigrams position-specific scoring matrix.

**Figure 4 ijms-25-04507-f004:**
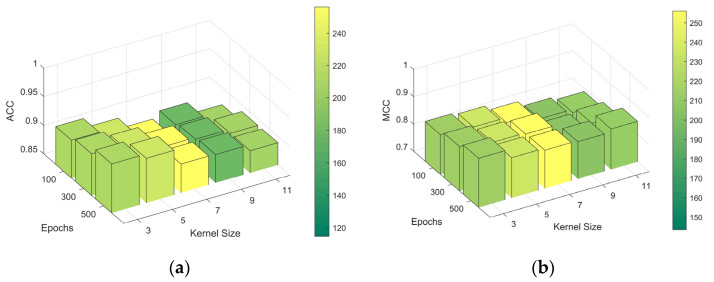
ACC and MCC plots of CapsNets with various kernel sizes across different training epochs. (**a**) ACC; (**b**) MCC. Abbreviations: ACC: accuracy; MCC: Matthews Correlation Coefficient; CapsNets: capsule networks.

**Figure 5 ijms-25-04507-f005:**
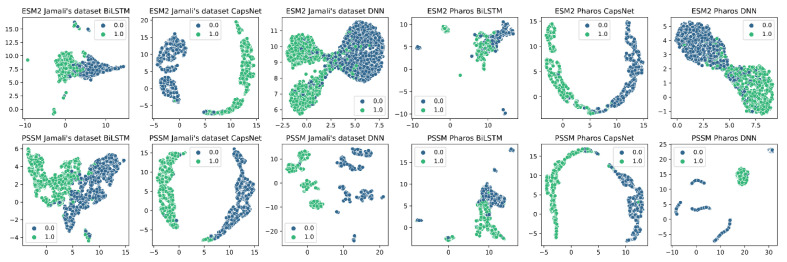
UMAP visualization on two datasets with different features and different models. Abbreviations: ESM2: evolutionary scale modeling 2; BiLSTM: bidirectional long short-term memory; CapsNet: capsule network; DNN: deep neural network.

**Figure 6 ijms-25-04507-f006:**
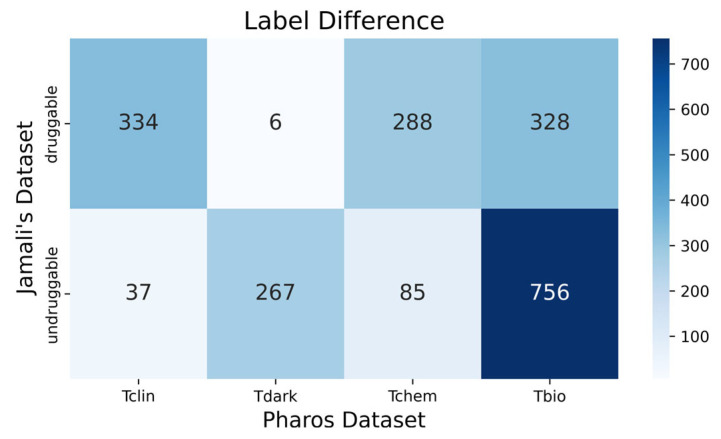
Heat map of labeling differences for the common protein sequences across these two datasets, totaling 2101 protein sequences.

**Figure 7 ijms-25-04507-f007:**
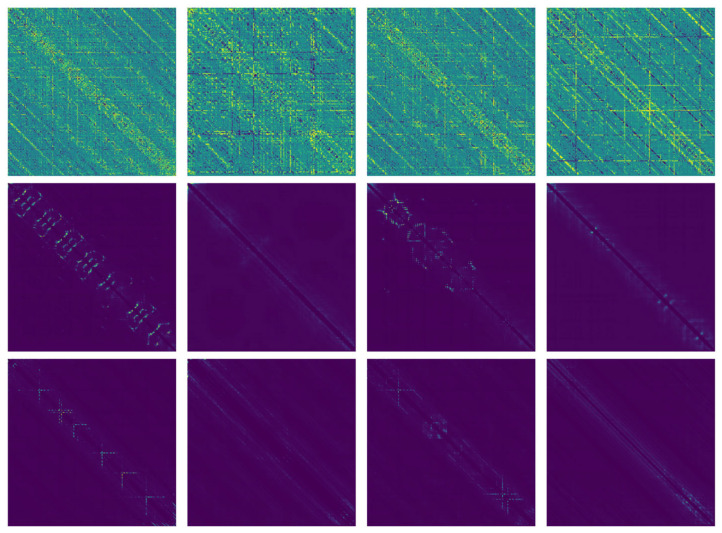
Protein contact maps. The first row is the underfitting ESM-2 model, the second row is the fine-tuned one on Jamali’s dataset, and the third row is the original ESM-2 model. For each row, it is labeled as undruggable protein (Jamali’s dataset), druggable protein (Jamali’s dataset), undruggable protein (Pharos dataset), and druggable protein (Pharos dataset). For each column, the protein is the same.

**Figure 8 ijms-25-04507-f008:**
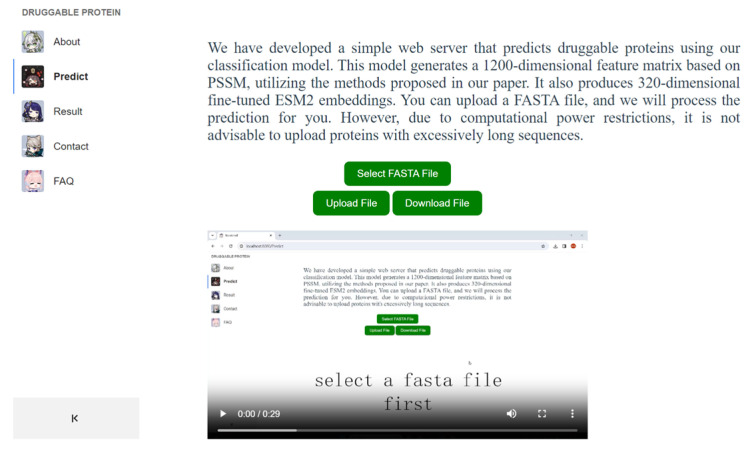
The web server interface.

**Figure 9 ijms-25-04507-f009:**
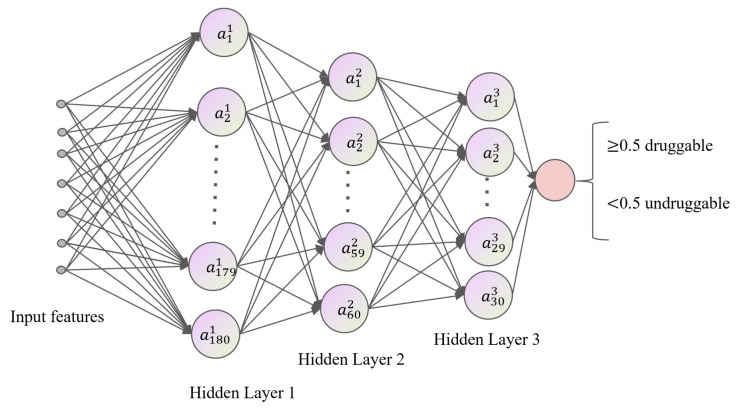
DNN architecture.

**Figure 10 ijms-25-04507-f010:**
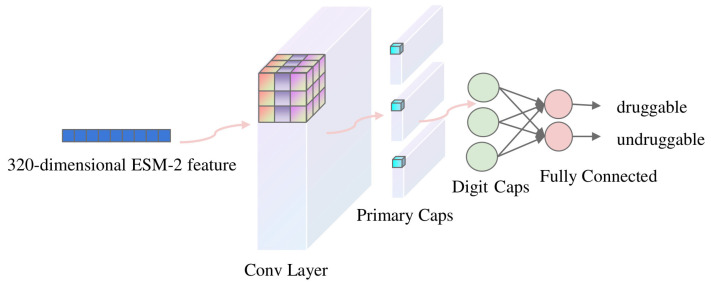
Capsule network architecture.

**Figure 11 ijms-25-04507-f011:**
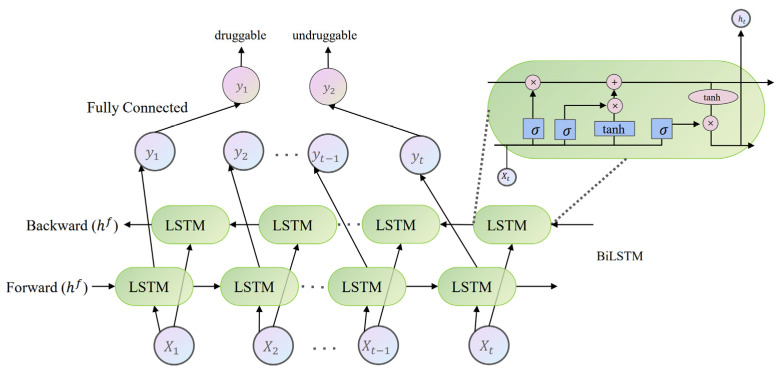
BiLSTM network architecture.

**Figure 12 ijms-25-04507-f012:**
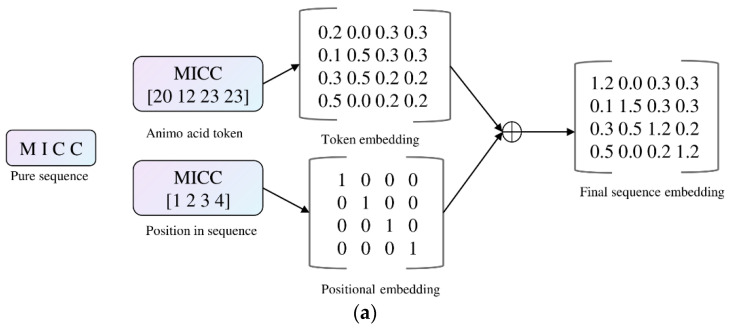
The modified GPT-2 for classification. (**a**) The embedding technique; (**b**) The modified architecture.

**Table 1 ijms-25-04507-t001:** Machine learning classifiers on Jamali’s dataset (test set).

Encoding	Classifier	ACC	P	SN	SP	F1	MCC
PSSM (1200 dimensions)	SVM	0.8271 ±0.0169	0.8280 ±0.0269	0.8214 ±0.0327	0.8326 ±0.0176	0.8247 ±0.0263	0.6541 ±0.0309
	XGB	0.9037 ±0.0245	0.9043 ±0.0203	0.9007 ±0.0332	0.9066 ±0.0146	0.9025 ±0.0200	0.8074 ±0.0212
	NB	0.7347 ±0.0295	0.8232 ±0.0503	0.5912 ±0.0646	0.8754 ±0.0256	0.6882 ±0.0471	0.4875 ±0.0597
	RF	0.8664 ±0.0269	0.8593 ±0.0341	0.8730 ±0.0220	0.8599 ±0.0169	0.8661 ±0.0268	0.7329 ±0.0276
ESM-2 (320 dimensions)	SVM	0.9326 ±0.0087	0.9406 ±0.0121	0.9196 ±0.0114	0.9449 ±0.0098	0.9300 ±0.0158	0.8652 ±0.0154
	XGB	0.9065 ±0.0170	0.8951 ±0.0179	0.9151 ±0.0246	0.8983 ±0.0114	0.9050 ±0.0289	0.8132 ±0.0369
	NB	0.8847 ±0.0153	0.8904 ±0.0156	0.8705 ±0.0247	0.8983 ±0.0250	0.8803 ±0.0258	0.7694 ±0.0289
	RF	0.8913 ±0.0177	0.8990 ±0.0237	0.8750 ±0.0331	0.9067 ±0.0127	0.8868 ±0.0135	0.7825 ±0.0318

Abbreviations: SVM: support vector machine; XGB: extreme gradient boosting; NB: naive bayes; RF: random forest; ACC: accuracy; P: precision; SN: sensitivity; SP: specificity; F1: F1 Score; MCC: Matthews Correlation Coefficient; PSSM: position-specific scoring matrix; ESM-2: evolutionary scale modeling 2.

**Table 2 ijms-25-04507-t002:** Machine learning classifiers on Pharos dataset (test set).

Encoding	Classifier	ACC	P	SN	SP	F1	MCC
PSSM (1200 dimensions)	SVM	0.8156 ±0.0296	0.8527 ±0.0287	0.7692 ±0.0326	0.8633 ±0.0135	0.8088 ±0.0311	0.6347 ±0.0470
	XGB	0.8226 ±0.0254	0.8842 ±0.0198	0.7482 ±0.0343	0.8992 ±0.0121	0.8106 ±0.0212	0.6540 ±0.0429
	NB	0.7234 ±0.0356	0.7731 ±0.0301	0.6433 ±0.0431	0.8057 ±0.0198	0.7022 ±0.0232	0.4546 ±0.0553
	RF	0.8120 ±0.0155	0.8947 ±0.0217	0.7132 ±0.0236	0.9136 ±0.0292	0.7937 ±0.0252	0.6387 ±0.0370
ESM-2 (320 dimensions)	SVM	0.8455 ±0.0285	0.8264 ±0.0352	0.8547 ±0.0126	0.8372 ±0.0194	0.8403 ±0.0356	0.6911 ±0.0419
	XGB	0.8739 ±0.0192	0.8208 ±0.0148	0.9401 ±0.0150	0.8139 ±0.0236	0.8764 ±0.0142	0.7562 ±0.0359
	NB	0.7195 ±0.0257	0.6578 ±0.0411	0.8547 ±0.0266	0.5968 ±0.0303	0.7434 ±0.0325	0.4641 ±0.0606
	RF	0.8739 ±0.0211	0.8307 ±0.0157	0.9207 ±0.0305	0.8294 ±0.0359	0.8749 ±0.0167	0.7528 ±0.0227

Abbreviations: SVM: support vector machine; XGB: extreme gradient boosting; NB: naive bayes; RF: random forest; ACC: accuracy; P: precision; SN: sensitivity; SP: specificity; F1: F1 Score; MCC: Matthews Correlation Coefficient; PSSM: position-specific scoring matrix; ESM-2: evolutionary scale modeling 2.

**Table 3 ijms-25-04507-t003:** Deep learning classifiers on Jamali’s dataset (test set).

Encoding	Classifier	ACC	P	SN	SP	F1	MCC
PSSM (1200 dimensions)	DNN	0.8827 ±0.0134	0.9318 ±0.0162	0.8233 ±0.0139	0.9409 ±0.0198	0.8742 ±0.0271	0.7703 ±0.0347
	CapsNet	0.8740 ±0.0173	0.8245 ±0.0284	0.9495 ±0.0147	0.7989 ±0.0277	0.8826 ±0.0137	0.7568 ±0.0276
	BiLSTM	0.8867 ±0.0144	0.9138 ±0.0167	0.8514 ±0.0229	0.9213 ±0.0226	0.8815 ±0.0125	0.7750 ±0.0304
ESM-2 (320 dimensions)	DNN	0.9105 ±0.0160	0.8991 ±0.0141	0.9145 ±0.0153	0.9069 ±0.0297	0.9067 ±0.0220	0.8209 ±0.0285
	CapsNet	0.9340 ±0.0129	0.9128 ±0.0312	0.9544 ±0.0164	0.9151 ±0.0119	0.9331 ±0.0211	0.8690 ±0.0252
	BiLSTM	0.8984 ±0.0171	0.8736 ±0.0184	0.9209 ±0.0131	0.8778 ±0.0172	0.8966 ±0.0219	0.7979 ±0.0278

Abbreviations: DNN: deep neural network; CapsNet: capsule network; BiLSTM: bidirectional long short-term memory; ACC: accuracy; P: precision; SN: sensitivity; SP: specificity; F1: F1 Score; MCC: Matthews Correlation Coefficient; PSSM: position-specific scoring matrix; ESM-2: evolutionary scale modeling 2.

**Table 4 ijms-25-04507-t004:** Deep learning classifiers on Pharos dataset (test set).

Encoding	Classifier	ACC	P	SN	SP	F1	MCC
PSSM (1200 dimensions)	DNN	0.8886 ±0.0120	0.8789 ±0.0218	0.9074 ±0.0197	0.8689 ±0.0319	0.8929 ±0.0317	0.7774 ±0.0293
	CapsNet	0.8602 ±0.0193	0.9026 ±0.0146	0.8148 ±0.0246	0.9078 ±0.0177	0.8564 ±0.0119	0.7245 ±0.0362
	BiLSTM	0.8578 ±0.0137	0.8333 ±0.0179	0.9028 ±0.0253	0.8107 ±0.0214	0.8667 ±0.0198	0.7175 ±0.0272
ESM-2 (320 dimensions)	DNN	0.9037 ±0.0164	0.8889 ±0.0243	0.9119 ±0.0114	0.8962 ±0.0166	0.9003 ±0.0226	0.8075 ±0.0203
	CapsNet	0.8997 ±0.0156	0.8677 ±0.0254	0.9318 ±0.0171	0.8705 ±0.0178	0.8986 ±0.0170	0.8017 ±0.0206
	BiLSTM	0.8222 ±0.0317	0.8870 ±0.0265	0.7185 ±0.0382	0.9167 ±0.0445	0.7939 ±0.0417	0.6516 ±0.0501

Abbreviations: DNN: deep neural network; CapsNet: capsule network; BiLSTM: bidirectional long short-term memory; ACC: accuracy; P: precision; SN: sensitivity; SP: specificity; F1: F1 Score; MCC: Matthews Correlation Coefficient; PSSM: position-specific scoring matrix; ESM-2: evolutionary scale modeling 2.

**Table 5 ijms-25-04507-t005:** Performance comparison with state-of-the-art models.

Model	ACC	SN	SP	F1	MCC
DrugMiner [10]	0.9210	0.9280	0.9134	0.9241	0.8417
GA-Bagging-SVM [11]	0.9378	0.9286	0.9445	0.9358	0.8781
XGB-DrugPred [30]	0.9486	0.9375	0.9574	0.9417	0.8900
DrugFinder [31]	0.9498	0.9633	0.9683	0.9460	0.8996
Modified GPT-2	0.9282	0.9332	0.9224	0.9332	0.8556
Fine-tunned ESM-2 with CapsNet	0.9511	0.9683	0.9691	0.9512	0.9011

Abbreviations: ACC: accuracy; SN: sensitivity; SP: specificity; F1: F1 Score; MCC: Matthews Correlation Coefficient; GPT-2: generative pre-trained transformer 2; GA-Bagging-SVM: high-level abbreviations for the model proposed in the original article; GA: Genetic Algorithm; Bagging: Bootstrap Aggregating; SVM: Support Vector Machine; XGB-DrugPred: high level abbreviations for the model proposed in the original article; XGB: eXtreme Gradient Boosting; DrugPred: Drug Prediction.

**Table 6 ijms-25-04507-t006:** Available web servers on druggable protein research.

Web Server Link	Summary	Data Available
https://www.drugminer.org	Search for druggable protein and view their features	yes
http://drugnomeai.public.cgr.astrazeneca.com	Provide visualization and clear explanation of the findings	yes
https://druggableprotein.com (ours)	Upload files in FASTA format and receive predicted results	yes

**Table 7 ijms-25-04507-t007:** Details of the samples in each process.

Dataset	Train Positive	Train Negative	Validation Positive	Validation Negative	Test Positive	Test Negative
Jamali’s	784	845	196	211	244	263
Pharos	452	452	112	112	140	140

**Table 8 ijms-25-04507-t008:** Summary of the data.

Dataset	Longest	Shortest	Mean	Medium	Standard Deviation
Jamali’s	5762	8	506	390	521
Pharos	34,350	2	554	411	528

**Table 9 ijms-25-04507-t009:** Final hyperparameters for machine learning models.

Model	Final Hyperparameters
SVM	C = 10, gamma = ’scale’, decision_function_shape = ’ovr’, kernel = ’rbf’
RF	n_estimators = 1000, max_depth = 3, random_state = 0, n_jobs = −1
NB	priors = None, var_smoothing = 1 × 10^−9^
XGB	max_depth = 15, learning_rate = 0.1, n_estimators = 2000, min_child_weight = 5, max_delta_step = 0, subsample = 0.8, colsample_bytree = 0.7, reg_alpha = 0, reg_lambda = 0.4, scale_pos_weight = 0.8, objective = ’binary:logistic’, eval_metric = ’auc’, seed = 1440, gamma = 0

Abbreviations: SVM: support vector machine; XGB: extreme gradient boosting; NB: naive bayes; RF: random forest.

## Data Availability

The data and the source code used to support the findings of this study are freely available to the academic community at https://github.com/txz32102/druggableprotein (accessed on 19 March 2024). The web server can be accessed by https://druggableprotein.com (accessed on 19 March 2024).

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
