# Peer review of "Comprehensive Research on Druggable Proteins: From PSSM to Pre-Trained Language Models"

_ijms, 2024, doi:10.3390/ijms25084507_

Round 1
Reviewer 1 Report
Comments and Suggestions for Authors
Brief Summary:
This manuscript provides an insightful exploration into the use of machine learning for predicting druggable proteins, employing various feature embedding techniques and deep learning algorithms. By experimenting with different combinations of methods on two datasets, the study illustrates the efficacy of computational models, particularly highlighting the performance of ESM-2 with CapsNet. This research could represent a fast and cost-efficient alternative to traditional experimental methodologies in drug discovery
However, I have several concerns that require clarification:
-
The process for preparing the data, specifically the division of training and test datasets, is unclear. Please provide more details on data split and how you utilized the 5-fold cross-validation for model selection and overfitting prevention.
-
The analysis of training outcomes seems to lack precision. Given the stochastic nature of deep learning models, determining the superiority of one model based solely on a single training outcome may not be robust. A confidence interval for the performance metrics might provide a more reliable assessment.
-
The abstract mentions an experiment utilizing GPT-2 for druggable protein prediction, yet the results indicate that CapsNet outperforms GPT-2. It appears that the contribution of GPT-2 to this paper may not be as pivotal as suggested.
-
The discussion on SHAP explanations appears quite brief. Have you performed a comparison among those features identified by different methodologies? Is there any inconsistency between the SHAP explanation and the existing findings?
-
The selection of hyperparameters for the DNN, CapsNet, and BiLSTM models is not adequately explained, raising questions about how these choices may have influenced the results.
Minor issues also noted include a typographical error: "CapsNet" should be correctly spelled in Tables 3 and 4.
Author Response
We greatly appreciate your thoughtful comments and constructive suggestions, which have significantly contributed to enhancing the quality of our manuscript. Further details have been outlined in the attached Word document. We hope these revisions adequately address your comments. We remain open to making further changes if necessary and look forward to your feedback.

Reviewer 2 Report
Comments and Suggestions for Authors
Authors presented an interesting and novel study of potential application of deep leaning methods to identify druggable proteins from their sequence information. Although this work is well written and potentially useful, it suffers from poor description of the datasets used in this analysis and a lack of clarity in describing algorithmic details.
Comments:
Line 84 - better embeddings in what sense- how the "better" is quantified?
Table1 and Table2 and Table3 and Table4 please explain the column names in the table's captions for consistency and to help a reader, regardless that you explained them in methods.
Please describe briefly Jamali's and Pharos datasets - what they consist of , what is the training data, number of features, sizes of training and test datasets at the first instance you mention them.
Line 101 , 109 on SHAP analysis and Figure 3- Please explain how from the representation in Figure 3 it is clear that S-FPSSM is the top feature? how the SHAP analysis is/should be interpreted in this context? What is a meaning of the magnitude of the SHAP value?
Lines 74-93 you refer to your research in the introduction as a work that is completed and that has led into the further investigation, but you do not cite any of your contributions, just references to the tools and methods created by others. Where are citations of YOUR work in introduction that you describe especially lines 87-93? If there is no reference to the mentioned work that completed by you, this must be research that you are presenting and it must go into Methods or the Results. Figure 1 caption must explain what is being represented by this Figure.
In Figure 5 - what are parameters of UMAP in these plots? Line 153 "high-dimensional data from the second-to-last layer" - what does it mean exactly and what is the purpose of Figure 5 at all? What do you want to show? Please explain it to the reader. What we must expect looking at these plots? What information they intend to convey?
Paragraph 2.3 and Figure 6. Better effort is required to explain the message of this paragraph 2.3. If you are comparing how many mistakes the methods made on Jamali and Pharos datasets, then the way you present it - by a confusion matrix- does not make any sense. Are the identities of the proteins same in these datasets? If yes, and you use confusion matrix to show the proportion of the mistakes, then you must explain it clearly. Otherwise, this confusion matrix presentation does not make any sense and you have to consider different way to communicate the ideas in 2.3.
You must do a better job explaining Figure 7. How these images represent "protein contact maps" line 191, lines 285-286? What is a "protein contact map" - please provide a definition and how these images in Figure 7 reflect that definition. Should not the protein contact map be represented by the amino acid sequence or a molecule graph?
Paragraph 3.1 Datasets. Please explain what elements in both datasets represent a protein- how the proteins are described - are they fully represented by sequences? What are feature vectors representing proteins? If the features are sequences, then please provide a summary statistics of the sizes of those sequences. Please clarify your selections of positive and negative protein examples in a Table - how many and from which source. The description how you formed a dataset from the Tbio, Tchem, Tdark, Tclin classes and the logic behind it is very confusing. It is not clear which Method 1 or 2 and how many data items items were used to present your findings in the current study.
Lines 324-325 How exactly "SHAP leverages game theory principles to quantitatively evaluate the influence of various input features on prediction outcomes." ? Please explain briefly.
Lines 331-332 - what parameters where used in final models , please provide a table with the values.
The BiLSTM architecture in Figure 11 is not complete. Please add clearly the output layer and the type of the output layer.
Author Response

(The authors gave the same response as above.)

Round 2
Reviewer 2 Report
Comments and Suggestions for Authors
Authors addressed all my concerns. Thank you for your interesting work.